# *Leishmania* Infection in Wild Lagomorphs and Domestic Dogs in North-East Spain

**DOI:** 10.3390/ani14071080

**Published:** 2024-04-02

**Authors:** Oscar Cabezón, Pamela Martínez-Orellana, Maria Puig Ribas, Catarina Jota Baptista, Diana Gassó, Roser Velarde, Xavier Fernández Aguilar, Laia Solano-Gallego

**Affiliations:** 1Wildlife Conservation Medicine Research Group (WildCoM), Departament de Medicina i Cirurgia Animals, Universitat Autònoma de Barcelona, 08193 Bellaterra, Catalonia, Spain; oscar.cabezon@uab.cat (O.C.); mariapuigribas@gmail.com (M.P.R.); xfdezaguilar@gmail.com (X.F.A.); 2Unitat Mixta d’Investigació IRTA-UAB, Centre de Recerca en Sanitat Animal (CReSA), Campus de la Universitat Autònoma de Barcelona (UAB), 08193 Bellaterra, Catalonia, Spain; 3Infectious and Inflammatory Diseases in Companion Animals Research Group (MIAC), Departament de Medicina i Cirurgia Animals, Universitat Autònoma de Barcelona, 08193 Bellaterra, Catalonia, Spain; pamela.martinez.phd@gmail.com; 4Egas Moniz Center for Interdisciplinary Research (CiiEM), Egas Moniz School of Health & Science, 2829-511 Almada, Portugal; catabap@hotmail.com; 5Departament de Ciència Animal, Escola Tècnica Superior d’Enginyeria Agroalimentaria i Forestal i de Veterinària (ETSEAFIV), Universitat de Lleida (UdL), 25199 Lleida, Catalonia, Spain; diana.gasso@udl.cat; 6Wildlife Ecology and Health Group (WE&H), and Servei d’ Ecopatologia de Fauna Salvatge (SEFaS), Departament de Medicina i Cirurgia Animals, Universitat Autònoma de Barcelona, 08193 Bellaterra, Catalonia, Spain; roser.velarde@uab.cat

**Keywords:** *Canis familiaris*, Catalonia, leishmaniosis, *Leishmania infantum*, *Lepus europaeus*, *Oryctolagus cuniculus*, seroprevalence, wildlife

## Abstract

**Simple Summary:**

*Leishmania infantum* is a zoonotic protozoan parasite transmitted by phlebotomine sandflies. Dogs are the main reservoir for human infections. In recent years, outbreaks of human leishmaniasis have been reported in different regions of Spain associated with the Iberian hare and European rabbit. However, there is a notable scarcity of information regarding *L*. *infantum* infection in the European hare and in Northeastern Spain where this species occurs. The present study aimed to assess *Leishmania* spp. exposure and infection in lagomorphs and sympatric domestic dogs in NE Spain. Results suggest a more important role for the European rabbit than the European hare in the epidemiology of this parasite in NE Spain. Given the strong correlation between lagomorph densities and human leishmaniasis outbreaks in Spain, the high rabbit and human densities in NE Spain, and the high *Leishmania* spp. seroprevalence in rabbits, it becomes relevant to establish surveillance programs for lagomorphs in this region.

**Abstract:**

*Leishmania infantum* is a zoonotic protozoan parasite distributed worldwide that is transmitted by phlebotomine sandflies. Dogs are the main reservoir for human infections. However, in recent years, the capacity of lagomorphs to contribute to *Leishmania* transmission has been confirmed. The present study aimed to assess *Leishmania* spp. exposure and infection in lagomorphs and sympatric domestic dogs in NE Spain. Sera from European hares, European rabbits, and rural dogs were tested for antibodies against *L. infantum* using an in-house indirect ELISA. PCR analysis targeting *Leishmania* spp. was performed in spleens from *L. europaeus*. Antibodies against *Leishmania* spp. were detected in all the species analyzed. Total sample prevalence was significantly higher in *O. cuniculus* (27.9%) than in *L. europaeus* (2.0%). Results of the PCR were all negative. The present study expands knowledge about *Leishmania* infections in free-ranging lagomorphs in the Iberian Peninsula, suggesting a more important role of *O. cuniculus* in the study area. Given the strong correlation between lagomorph densities and human leishmaniasis outbreaks in Spain, the high rabbit and human densities in NE Spain, and the high *Leishmania* spp. seroprevalence in rabbits, it becomes imperative to establish surveillance programs for lagomorphs in this region.

## 1. Introduction

*Leishmania infantum* (Kinetoplastida: Trypanosomatidae; Nicolle, 1908) is a zoonotic protozoan parasite distributed worldwide that is transmitted by phlebotomine sandflies [1]. In Mediterranean countries, leishmaniasis is reported yearly with a clinical disease ratio of 0.02–0.49/100,000 in people [2,3]. Several different species of *Leishmania* have been described that cause disease in animals and humans in the Mediterranean area: *L. infantum* is the most frequent and it is considered endemic in the whole area; *L. donovani*, Laveran and Mesnil, 1903, occurs in Cyprus; *L. major*, Yakimoff and Schokhor, 1914, occurs in North Africa and the Middle East; and *L. tropica*, Wright, 1903, occurs in Greece, Turkey, the Middle East, and North Africa [4]. Dogs are the main reservoir of *L. infantum* for human infections, with a prevalence of infection in this species ranging between 5% and 50% in Spain [5,6]. However, *L. infantum* has also been detected in different wildlife species, such as rodents, carnivores, and lagomorphs in Europe [7]. Recently, there has been documented evidence of sylvatic cycles, shedding light on the increasing importance of wildlife reservoirs of *L. infantum* in the Mediterranean basin [7,8]. The expansion of urbanised areas into natural habitats, and the increase of urban wildlife in green zones and parks in populated areas may be also promoting the abundance of *Phlebotomus* sandflies and leishmaniasis [9,10].

The capacity of lagomorphs to contribute to *Leishmania* transmission has been confirmed by xenodiagnoses. Among the four lagomorph species naturally occurring in Spain, the European hare (*Lepus europaeus*; Pallas, 1778), the Broom hare (*Lepus castroviejoi*; Palacios, 1976) the Iberian hare (*L. granatensis*; Rosenhauer, 1856), and the European rabbit (*O. cuniculus;* Linnaeus, 1758), the transmission of *L. infantum* back to sandflies has been confirmed in the latter two [10,11,12]. In addition, outbreaks of human leishmaniasis have been reported in different regions of Spain associated with lagomorphs. Between 2009 and 2012, a total of 449 visceral and cutaneous leishmaniasis human cases, representing a ratio of 56 cases of leishmaniasis per 100,000 human beings, were recorded in Central Spain [11]. In this outbreak, a sylvatic cycle maintained by lagomorphs was confirmed as the main source of the human cases, which were mostly associated with visits to an urban park. In that park, *L. granatensis* were at high densities and were considered the primary reservoir species of *L. infantum*, potentially with the added contribution of *O. cuniculus* [11,13]. Similarly, in Granada province in South Spain, an increase in the incidence of human leishmaniasis between 2008 (0 cases) and 2016 (11.3 per 100,000 inhabitants) was associated with the overlapping of domestic and sylvatic cycles, and *O. cuniculus* was considered the main reservoir species [14].

Some authors have pointed out the usefulness of determining the prevalence of infection and parasite burden in lagomorphs to control and reduce leishmaniasis risk to people [14]. Considering the significance of lagomorphs in the spread of *L. infantum* in Spain, numerous epidemiological studies have focused on these species, demonstrating high exposure and infection prevalence [14,15,16,17,18]. Nevertheless, there is a notable scarcity of information regarding *L*. *infantum* infection in the European hare and in Northeastern Spain where this species occurs. 

The Iberian Peninsula is one of the most biodiverse areas in Europe, with highly diverse ecoregions. NE Spain (Western Mediterranean basin) is characterized by hot and dry summers and relatively temperate and humid-to-sub-humid winters. The annual average temperature ranges from 10 to 17 °C, and the minimum average temperature of the coldest month ranges from 5 to 10 °C. The annual precipitation ranges from 350 to 800 mm, typified by torrential rainfall in autumn [19]. In this ecoregion, *L. infantum* is found endemic in domestic dogs [20].

The present study aimed to assess *L. infantum* exposure and infection in European hares from Northeastern Spain. We also analyzed samples from sympatric rabbits and domestic rural dogs, which are the two other competent hosts known to have significant contributions to *L. infantum* epidemiology in NE Spain [15,20].

## 2. Materials and Methods

### 2.1. Animals and Samples

Blood (n = 147) and spleen (n = 20) samples were obtained from 158 hunted *L. europaeus* from three different provinces within NE Spain: Girona (n = 122), Lleida (n = 14), and Barcelona (n = 11) (Figure 1). All hares were sampled from October to March between 2012 and 2017 during the hunting season. There were 66 females and 60 males, and 32 had unknown gender. Also, blood samples from 111 *O. cuniculus* were collected between 2014 and 2017 in Lleida (n = 54), Tarragona (n = 13), Girona (n = 3), and Barcelona (n = 37), and four more were sampled without a known location within the study area. Rabbits were sampled during the regular hunting season (October to February) and included 46 females, 31 males, and 34 rabbits with unknown gender. Finally, sera were obtained from 226 rural dogs (*Canis familiaris*) from Girona (n = 115), Lleida (n = 39), Barcelona (n = 7), and Tarragona (n = 65) between 2012 and 2016. In all cases, the dogs were housed in outdoor facilities and were mostly used for hunting activities, livestock rearing, or as guard dogs.

### 2.2. Serological and Molecular Analyses

All sera from *L. europaeus*, *O. cuniculus*, and dogs were tested for antibodies against *L. infantum* using an in-house indirect ELISA previously described [19]. This technique was modified for sera samples of *Lagomorpha* species. Sera were diluted to 1:400 in hare and rabbit samples and to 1:800 in dog samples, and incubated in sonicated crude *L. infantum* antigen-coated plates (20 μg/mL) for 1 h at 37 °C. The plates were washed with 0.05% Tween 20 in phosphate-buffered saline (PBS) and incubated with Protein A conjugated to horseradish peroxidase (1:30,000 dilution; Sigma-Aldrich, Madrid, Spain) for 1 h at 37 °C. Plates were rewashed with 0.05% PBS–Tween 20. Subsequently, substrate solution ortho-phenylenediamine (OPD) and stable peroxide substrate buffer (Sigma-Aldrich, Madrid, Spain) were added. The reaction was stopped with 50 μL of 2.5 M H_2_SO_4_. Absorbance values were read at 492 nm in an automatic micro-ELISA reader (ELISA Reader Anthos 2020, Biochrom, Cambridge, UK). All plates included the serum from two infected animals with a confirmed infection (hare controls in lagomorph assays and dog controls in dog assays), one of which was the positive control (calibrator), and serum from a non-infected hare, non-infected rabbit, and non-infected dog were used as negative controls. All samples were analysed in duplicate. Results were quantified as ELISA units (EU) in relation to the controls that were used as calibrators, and arbitrarily set at 100 EU in dogs, and lagomorphs were set at 170 EU. 

The cut-offs were established by different methods between the two lagomorph species. For *L. europaeus*, the cut-off was established at 2.43 EU for negative samples and at 4.14 EU for positive samples (calculated as the mean + 2 SD and mean + 4 SD, from values of 29 hares from non-endemic areas). For *O. cuniculus*, the cut-off was established at 2.0 EU for negative samples and at 2.8 EU for positive samples (calculated as the mean + 4 SD and mean + 6 SD, respectively, from values of 15 *O. cuniculus* SPF (some pathogens free)). Sera from hares were classified as positive when ≥4.14 EU, doubtful when ≥2.43 EU and <4.14 EU, and negative with EU < 2.43 EU. Sera from *O. cuniculus* were classified as positive when ≥2.8 EU, doubtful when ≥2.0 EU and <2.8 EU, and negative when <2.0 EU. The cut-off for dog samples was established at 35 EU (mean + 4 SD from values of 80 dogs from non-endemic areas, United Kingdom). 

PCR analysis targeting *Leishmania* spp. was performed in 20 spleen samples from European hares. DNA was extracted from spleen homogenates using the commercial kit BioSprint 96 DNA Blood Kit (QIAGEN, Hilden, Germany), according to the manufacturer’s procedure, and the extraction robot Biosprint 96 (QIAGEN, Hilden, Germany). Extracted DNA was amplified using real-time-PCR with the primers Leish-1 (3′ AACTTTTCTCGTCCTCCGGGTAG 5′) and Leish-2 (3′ ACCCCCAGTTTCCCGCC 5′) [21], the probe (3′ FAM-AAAAATGGGTGCAGAAAT-MGB 5′), and a commercial kit (TaqMan PCR Master Mix; Applied Biosystems, Carlsbad, CA, USA).

### 2.3. Review of Leishmania Infection in Wild Lagomorphs in Spain

A review of *Leishmania* spp. infection in wild lagomorphs (*O. cuniculus*, *L. europaeus*, *L. castroviejoi*, and *L. granatensis*) in Spain was performed using a systematic search and compilation methodology of available peer-reviewed literature. We searched Web of Science: All Databases (WoS; Thomson Reuters, Toronto, ON, Canada) literature database using “topic” searcher. We used the words “(*Leishmania* AND lagomorph AND Spain)”, and then we selected epidemiological studies on *L. infantum* infection in wild lagomorphs in Spain. 

### 2.4. Statistical Analysis

Differences in antibody prevalence between species, geographic locations (provinces within NE Spain), and sex of the animals were tested with a Pearson’s chi-squared test or Fisher’s exact test when one or more cells in the contingency table had less than five cases. All analyses were performed using R software, version 4.1.0, and the 95% interval confidence of apparent prevalence was calculated with the “EpiR” package 2.0.61 [22]. The significance level was set at 0.05 in all tests.

## 3. Results

### 3.1. Serological and Molecular Analyses

Antibodies against *L. infantum* antigen were detected in all the species analyzed and in all provinces in at least one of the species. Remarkably, positive cases of *L. europaeus* were only detected in Girona province, which is the region with a lower sample prevalence in *O. cuniculus* and dogs (Table 1). The results of the serology, considering the different species and geographical locations, are summarized in Table 1.

The differences in sample prevalence between species were statistically significant (Fisher’s exact test *p*-value < 0.001). Among lagomorphs, total sample prevalence was also significantly higher in *O. cuniculus* than in *L. europaeus* (Fisher’s exact test *p*-value < 0.001; OR 18.5; 95% CI: 6.3–82.2). There were no statistical differences between males and females for all three species. Regarding geographical locations (provinces within NE Spain), sample prevalence was higher in Tarragona for *C. familiaris* (Fisher’s exact test *p*-value = 0.002; OR 1.8; 95% CI: 0.3–48.0) but not for *L. europaeus* or *O. cuniculus*, in which sample prevalence did not significantly vary between provinces. 

Results of the PCR analysis for the detection of *L. infantum* in *L. europaeus* spleens were all negative (0.0%; 95% CI: 0.0–16.1).

### 3.2. Leishmania infantum in Lagomorphs in Spain

A literature review found nine original research articles reporting epidemiological studies of *L. infantum* infection in lagomorphs in Spain. A summary of the studies is presented in Table 2. The most studied lagomorph species studied were *L. granatensis* and *O. cuniculus.* Only one study included samples of *L. europaeus*, with a few individuals analyzed (n = 14). 

## 4. Discussion

The sample prevalence of antibodies against *L. infantum* found in the present study confirms exposure in all the species sampled, further indicating the endemic status of *Leishmania* in NE Spain and the Mediterranean basin. *Oryctolagus cuniculus* had a significantly higher sample prevalence as compared to *L. europaeus* and domestic dogs, suggesting a more important role of this species in the epidemiology of the parasite in the study area. This is in accordance with previous studies reporting high exposure and infections of *Leishmania* in lagomorphs in Spain (Table 2).

These previous studies found a high variability of *Leishmania* infection in lagomorphs, and several factors have been proposed to drive these differences (e.g., host distribution, host density, vector density, and host susceptibility). In our area of study, the geographic distributions of species sampled (i.e., *L. europaeus*, *O. cuniculus*, and domestic dog) overlap [26,27], and the presence of sandflies and the parasite are known to occur throughout the ecoregion. However, the overall density of wild rabbits (29–300 individuals/km^2^) is higher than the density of *L. europaeus* (1.29–13 individuals/km^2^) and domestic dogs [26,27]. Also, the ecology of *O. cuniculus* favors higher aggregations of individuals and the attraction of sandflies [10,28,29]. These high densities, together with the feeding preference of sandflies for lagomorphs [13,30,31], may increase the prevalence of *Leishmania* infection in *O. cuniculus*. Nonetheless, our study provides a snapshot overview of *L. infantum* epidemiology in North-East Spain, and dynamics of infection may vary locally or temporally depending on hosts and vector abundance.

*Leishmania* spp. are also widespread in other areas of the Mediterranean basin. However, *Leishmania* infection is underreported in lagomorphs in this area, and no studies on the potential role of lagomorphs in the epidemiology of human and animal leishmaniasis are available [4]. The few studies performed on *Leishmania* spp. infection in wild lagomorphs in other geographic areas from the Mediterranean basin point to a wide heterogeneity of epidemiological scenarios, including the circulation of different species of the parasite [32]. 

In the Mediterranean basin, *Leishmania* spp. infection has been reported in wild lagomorphs contrasting a wide range of results on both exposure and molecular detection of the parasite. In Tuscany, Central Italy, 9.8% (n = 51) of *L. europaeus* sampled were positive in PCR analyses [33]. Also, in Central Italy, in the province of Pisa, Ebani et al. [34] found that 0.9% (n = 222) of *L. europaeus* had antibodies against *Leishmania* spp. by IFAT. In North-East Italy, Taddei et al. [35] recorded that 15.4% (n = 39) of free-ranging European hares were positive for *Leishmania* DNA by PCR. In Northern Italy, Zanet et al. [36] found *Leishmania* DNA in 30% (CI95% 10.78–60.32%; n = 10) of *O. cuniculus* analyzed, in 26.92% (CI95% 19.33–36.16%; n = 104) of exotic invasive Eastern cottontails (*Sylvilagus floridanus*), and in 18.52% (CI95% 12.32–26.88%; n = 108) of *L. europaeus*. 

Abbate et al. [37] reported a positivity of 4.2% (n = 71) by qPCR in wild rabbits of Sicily, Southern Italy. In Greece, different prevalence of *Leishmania* infections have been documented in European hares [32]. In Northern Greece, a sample prevalence of 23.49% (n = 166) was found by PCR [38]. Also, Tsokana et al. [39] studied hares from Northern and Central Greece, detecting antibodies in 12.4% (n = 105) and *Leishmania* DNA in 9.6% (n = 52) of the hares. In the same geographic area, Tsakmakidis et al. [40] found a similar prevalence of antibodies (6.7%; n = 90) and *Leishmania* DNA (3.6%; n = 56) in hares. Furthermore, *Leishmania* spp. infection has been studied in *O. cuniculus* from Greece showing similar exposure to that reported in hares from the same country. Tsakmakidis et al. [40] observed a seroprevalence of antibodies of 7.6% (n = 393), and *Leishmania* DNA was detected in 2.6% (n = 113) of the rabbits from Northern Greece. 

The different *Leishmania* species circulating in the Eastern Mediterranean basin, the heterogeneity of *Leishmania* infection rates in lagomorphs, and the few studies on the role of these species in the epidemiology of *Leishmania* spp. in the Mediterranean basin reinforce the need for further research in lagomorphs. 

Wildlife species are important for the epidemiology and transmission of zoonotic diseases and sometimes can represent a reservoir of disease to people [41]. However, the diversity of pathogens that naturally circulate in wild populations, and the characteristics of the host populations that favour the emergence of the zoonoses are far from being understood [9,41]. Several studies have investigated how synanthropic animals may increase the risk of spill-over of zoonotic diseases from wildlife reservoirs to people [42,43]. In Spain, human leishmaniasis outbreaks have been related to high densities of the two proven *L. infantum* wildlife reservoirs, *L. granatensis* and *O. cuniculus*, when they were in proximity to densely human-populated areas [8,9,10,11]. Currently, NE Spain is home to more than 7.5 M human beings, representing 16.4% of the population of Spain [44]. Also, the populations of *O. cuniculus* in recent years in NE Spain have been increasing and are requiring management measures to control their populations [45]. Although no outbreaks of human leishmaniasis have been noticed in NE Spain to date, sporadic cases may have been underreported [4]. Given the identified strong correlation between lagomorph densities and human leishmaniasis outbreaks in Spain, the high rabbit and human densities in NE Spain, and the high *L. infantum* seroprevalence in rabbits, it becomes imperative to establish robust surveillance programs for lagomorphs in this ecoregion. Furthermore, the dynamics of *Leishmania* in lagomorph species is not well understood and, therefore, longitudinal studies are necessary. 

## 5. Conclusions

The present study expands the knowledge about *Leishmania* infections in free-ranging lagomorphs in the Iberian Peninsula and reinforces the hypothesis of the important role of *O. cuniculus* in the epidemiology of the parasite in North-East Spain. In contrast, the European hare does not seem to play an important role in the overall epidemiology of *Leishmania* spp. in this region.

## Figures and Tables

**Figure 1 animals-14-01080-f001:**
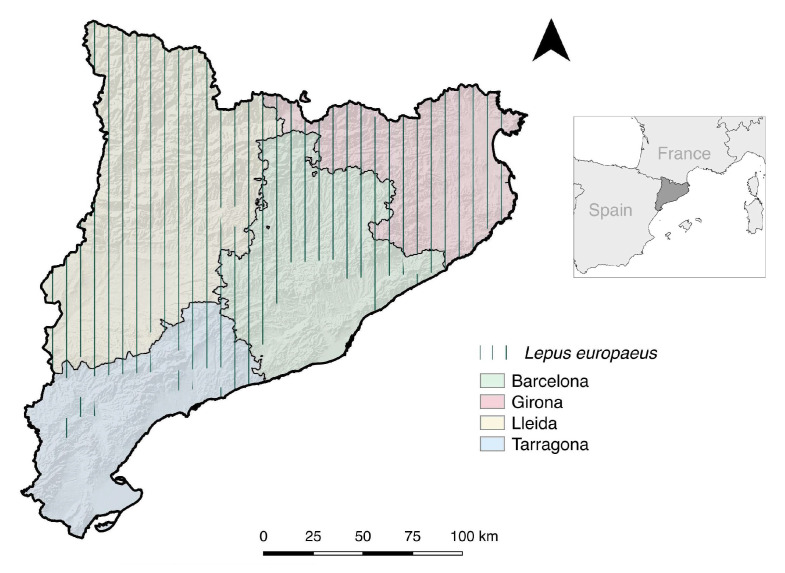
Map of the study area in Catalonia, NE Spain, indicating provinces sampled (Barcelona, Girona, Lleida, and Tarragona), and the distribution of the European hare (*Lepus europaeus*). The European rabbit (*Oryctolagus cuniculus*) is widely distributed throughout the territory in all provinces within Catalonia.

**Table 1 animals-14-01080-t001:** Apparent sample prevalence (%) of antibodies against *L. infantum* in European hare (*Lepues europaeus*), European rabbit (*O. cuniculus*), and rural domestic dogs in the four provinces of Northeastern Spain Mediterranean Ecoregion (Girona, Tarragona, Lleida, and Barcelona).

Species	Girona	Tarragona	Lleida	Barcelona	Total
	% (n)	% (n)	% (n)	% (n)	% (n; 95% CI)
*Lepus europaeus*	2.7 (122)	-	0 (14)	0 (11)	2.0 (147; 0.7–5.8)
*Oryctolagus cuniculus*	0 (3)	46.1 (13)	20.4 (54)	37.8 (37)	27.9 (111; 20.4–36.9) *
*Canis familiaris*	5.2 (115)	24.6 (65)	10.3 (39)	14.29 (7)	11.9 (226; 8.3–16.8)
Total	3.8 (240)	27.8 (78)	13.0 (107)	25.4 (55)	12.6 (484; 9.9–15.9)

* The total sample size of *O. cuniculus* does not sum up because four individuals were from unknown provinces within the study area. n: number of samples analyzed; CI: confidence intervals.

**Table 2 animals-14-01080-t002:** Prevalence of *L. infantum* in lagomorph species from Spain (i.e., European rabbit–*Oryctolagus cuniculus*; European hare–*Lepus europaeus*; Iberian hare–*Lepus granatensis*; Broom hare–*Lepus castroviejoi*) reported in the literature.

	Geographic Origin	% (Analyzed)	Method	Sample	Ref
*L. europaeus*	Mediterranean NE Spain	2.0% (147)	ELISA	Serum	Present study
		0% (20)	PCR	Spleen	
*O. cuniculus*	Catalonia, NE Spain	27.9% (111)	ELISA	Serum	
*L. europaeus*	Atlantic, Northern Spain	64.25% (14)	PCR	Spleen	[16]
	NE Spain	0% (2)	PCR	Spleen	
*L. castroviejoi*	Atlantic, Northern Spain	0% (2)	PCR	Spleen	
	Central Spain	60% (10)	PCR	Spleen	
*L. granatensis*	NE Spain	60% (5)	PCR	Spleen	
	Northern plateau	20% (5)	PCR	Spleen	
	Southern plateau	38.8% (54)	PCR	Spleen	
	Guadalquivir valley, South Spain	50% (2)	PCR	Spleen	
*L. granatensis*	Southern Madrid, Central Spain	74.1% (85) *	IFAT	Serum	[12]
*O. cuniculus*		45.7% (35) *	IFAT	Serum	
*O. cuniculus*	Mediterranean SE Spain	0% (36)	ELISA	Serum	[15]
		0.8% (129)	PCR	Spleen	
*O. cuniculus*	Loja, Granada, SE Spain	100% (40) *	PCR	Skin	[14]
	Huéscar, Granada, SE Spain	0% (11)	PCR	Skin	
*L. granatensis*	Northern Madrid, Central Spain	17.4% (69)	IFAT	Serum	[23]
		1.45% (69)	PCR	Skin	
*O. cuniculus*		49.8% (215)	IFAT	Serum	
		12.1% (215)	PCR	Skin	
*O. cuniculus*	South-East Spain	30% (80)	rtPCR	Skin	[17]
*L. granatensis*		0% (1)			
*O. cuniculus*	Granada, SE Spain	28.6% (150)	IFAT	Serum	[18]
		20.7% (150)	PCR	blood, liver, Spleen, heart, skin	
*O. cuniculus*	Madrid, Central Spain	75.4% (69) *	IFAT	Serum	[24]
		2.9%/17.4% (69) *	PCR	Spleen/Skin	
*L. granatensis*	Madrid, Central Spain	85.71% (7) *	DFA assay	Multiple	[25]
		14.29% (7) *	IFAT	Spleen exudate	
*O. cuniculus*		36.54% (52) *	DFA assay	Multiple	
		0% (50) *	IFAT	Spleen exudate	

L.: *Lepus*. O.: *Oryctolagus*. (*): animals from areas with emerging human leishmaniasis. IFAT: Indirect Fluorescent Antibody Technique. DFA assay: direct fluorescence antibody assay.

## Data Availability

The raw data supporting the conclusions of this article will be made available by the authors on request.

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
