# Peer review of "Leishmania Infection in Wild Lagomorphs and Domestic Dogs in North-East Spain"

_animals, 2024, doi:10.3390/ani14071080_

Round 1

Reviewer 1 Report

Comments and Suggestions for Authors

The manuscript entitled “Leishmania infantum infection in wild lagomorphs and domestic dogs in North-East Spain” by Cabezón and collaborators aims to determine the seroprevalence for leishmania in hares, rabbits and domestic dogs from the northeast region of Spain and to alert to the role of these species on the epidemiology of the disease.

Although Leishmania infantum antigen had been used in serological tests, this does not prove infection by this species of the parasite, since it is known that there is a strong cross-reactivity between all species of the genus Leishmania in serological tests, in addition to cross-reactivity to other infectious agents, which were not discarded. Added to this, no PCR was positive, material that could have been used to characterize the Leishmania species.

Even though we know the high prevalence of L. infantum infection in the region, I find it fearful to state in the conclusion of the study and even in the title of the manuscript that the infection in lagomorphs and dogs is by L. infantum infection. I believe it could be more prudent the authors conclude infection by Leishmania sp in lagomorphs and dogs in northeast of Spain and just suggest in the discussion that the infection could be by L. infantum based on the epidemiology of the region.

In general, the manuscript is very well written, the objectives are clear and well defined and certainly add new results to the area of knowledge. The methodology is written in details, the results are properly presented, illustrated, interpreted and discussed.

Author Response

Reviewer 1.1 (R1). Even though we know the high prevalence of L. infantum infection in the region, I find it fearful to state in the conclusion of the study and even in the title of the manuscript that the infection in lagomorphs and dogs is by L. infantum infection. I believe it could be more prudent the authors conclude infection by Leishmania sp in lagomorphs and dogs in northeast of Spain and just suggest in the discussion that the infection could be by L. infantum based on the epidemiology of the region.

Authors 1.1 (A1). This appreciation has been included in Title and the Discussion section of the manuscript.

Reviewer 2 Report

Comments and Suggestions for Authors

The manuscript is devoted to the Leishmania distribution in lagomorphs and dogs in North-Eastern Spain. Leishmaniasis is a zoonotic vector-borne disease, which are can transmitted to humans. Therefore, the study of Spanish colleagues for Leishmania infection is interesting and relevant to actual theme. The authors used serological and molecular analyzes of Leishmania on a large representative sample of blood and spleen from lagomorphs and dogs. A review of studies on Leishmania infection in other Mediterranean countries certainly enriched the article.

However, I have some comments about the manuscript.

1. Lines 95-101 – This paragraph looks like an extraneous in this section. It’s better to move the region description to the Introduction, and make it penultimate here.

2. I would suggest that the authors move Table 2 to the Results as a separate subsection with approximately the same title: “Distribution of Leishmania infantum in Lagomorphs in Spain”. line 189 – no literature references needed here. They are already in the table 2.

3. It is advisable to provide a schematic map of the study areas (or their geographic coordinates) in the Materials and Methods, because not everyone knows where it is.

4. According the International Code of Zoological Nomenclature (ICZN), at the first mention of species or genus in article text, its full Latin name with the author and year of description should be given. For instance: Leishmania infantum Nicolle, 1908 (Line 50), Leishmania donovani Laveran et Mesnil, 1903 (Line 55), etc.  For hosts (lagomorhs), it is also desirable at the first mention to give Latin names with author and year of description.

The verb “reported” appears 16 times in the text. Especially often in a newly written section. Probably you can replace some of them with synonyms: recorded, revealed, found…

5. In scientific articles, it is better to use only the Latin names of animals, avoiding the use of common names. Lepus europaeus is preferable to the European hare. The common name can be given at the first mention of the animal along with the Latin name.

Small remarks:

Line 54 - It is better to move reference [4] to the end of the sentence

Line 211- Now the word “however” is unnecessary here.

Lines 46,162,174, 177,180,209 – Latin names in italics.

Authors must submit the manuscript in accordance with the rules for formatting MDPI articles. Article title and the names of sections and subsections, all words in which must be capitalized (except for the parasite names).

I express my opinion – the manuscript can be published in Animals, but minor corrections are needed.

Author Response

Reviewer 2.1. (R.2.1.) Lines 95-101 – This paragraph looks like an extraneous in this section. It’s better to move the region description to the Introduction, and make it penultimate here.

Authors 2.1. (A.2.1.) Done. The paragraph has been moved to the Introduction section.

R.2.2. I would suggest that the authors move Table 2 to the Results as a separate subsection with approximately the same title: “Distribution of Leishmania infantum in Lagomorphs in Spain”. line 189 – no literature references needed here. They are already in the table 2.

A.2.2. Table 2 has moved to “Results” section. Also, a sub-section has been included in “Materials and Methods”.

R.2.3. It is advisable to provide a schematic map of the study areas (or their geographic coordinates) in the Materials and Methods, because not everyone knows where it is.

A.2.3. A map (Figure 1) has been included in the manuscript in the M&M section, detailing the location of Catalonia (NE Spain), the four provinces studied and the geographic distribution of European hare in the territory.

R.2.4. According the International Code of Zoological Nomenclature (ICZN), at the first mention of species or genus in article text, its full Latin name with the author and year of description should be given. For instance: Leishmania infantum Nicolle, 1908 (Line 50), Leishmania donovani Laveran et Mesnil, 1903 (Line 55), etc. For hosts (lagomorhs), it is also desirable at the first mention to give Latin names with author and year of description.

A.2.4. Done.

R.2.5. The verb “reported” appears 16 times in the text. Especially often in a newly written section. Probably you can replace some of them with synonyms: recorded, revealed, found…

A.2.4. Done

R.2.5. In scientific articles, it is better to use only the Latin names of animals, avoiding the use of common names. Lepus europaeus is preferable to the European hare. The common name can be given at the first mention of the animal along with the Latin name.

A.2.5. The common names of European hare and European rabbit have been replaced by their relative scientific names in most of cases within the manuscript.

Small remarks:

R.2.6. Line 54 - It is better to move reference [4] to the end of the sentence

A.2.6. Done.

R.2.7. Line 211- Now the word “however” is unnecessary here.

A.2.7. Done.

R.2.8. Lines 46,162,174, 177,180,209 – Latin names in italics.

A.2.8. Done. The manuscript has reviewed again to check all the scientific names (latin) are in italics.

R.2.9. Authors must submit the manuscript in accordance with the rules for formatting MDPI articles. Article title and the names of sections and subsections, all words in which must be capitalized (except for the parasite names).

A.2.9. This appreciation has been included in the manuscript.